# UGGT1-mediated reglucosylation of *N*-glycan competes with ER-associated degradation of unstable and misfolded glycoproteins

**Satoshi Ninagawa[1,2]\*[†], Masaki Matsuo[2][†], Deng Ying[3][†], Shuichiro Oshita[2], Shinya Aso[2], Kazutoshi Matsushita[3], Mai Taniguchi[3], Akane Fueki[2], Moe Yamashiro[2], Kaoru Sugasawa[1,4], Shunsuke Saito[3], Koshi Imami[5], Yasuhiko Kizuka[6], Tetsushi Sakuma[7], Takashi Yamamoto[7], Hirokazu Yagi[8,9], Koichi Kato[8,9,10], Kazutoshi Mori[3,11]\***

[1]Biosignal Research Center, Kobe University, Kobe, Japan; [2]Department of Bioresource Science, Graduate School of Agricultural Science, Kobe University, Kobe, Japan; [3]Department of Biophysics, Graduate School of Science, Kyoto University, Kyoto, Japan; [4]Graduate School of Science, Kobe University, Kobe, Japan; [5]Proteome Homeostasis Research Unit, RIKEN Center for Integrative Medical Sciences, Kanagawa, Japan; [6]Laboratory of Glycobiochemistry, Institute for Glyco-core Research (iGCORE), Gifu University, Gifu, Japan; [7]Division of Integrated Sciences for Life, Graduate School of Integrated Sciences for Life, Hiroshima University, Hiroshima, Japan; [8]Graduate School of Pharmaceutical Sciences, Nagoya City University, Nagoya, Japan; [9]Exploratory Research Center on Life and Living Systems (ExCELLS), National Institutes of Natural Sciences, Okazaki, Japan; [10]Institute for Molecular Science (IMS), National Institutes of Natural Sciences, Okazaki, Japan; [11]Institute for Advanced Study, Kyoto University, Kyoto, Japan

**\*For correspondence:**
sninagawa@harbor.kobe-u.ac.
jp (SN);
mori.kazutoshi.8r@kyoto-u.ac.
jp (KM)

[†]These authors contributed
equally to this work

**Competing interest:** The authors
declare that no competing
interests exist.

**Reviewing Editor:** Luke
Wiseman, Scripps Research
Institute, United States

## eLife Assessment

This **important** manuscript demonstrates that UGGT1 is involved in preventing the premature degradation of endoplasmic reticulum (ER) glycoproteins through the re-glucosylation of their *N*-linked glycans following release from the calnexin/calreticulin lectins. The authors include a wealth of **convincing** data in support of their findings, although extending these findings to other types of substrates, such as secreted proteins, could further demonstrate the global importance of this mechanism for protein trafficking through the secretory pathway. This work will be of interest to scientists interested in ER protein quality control, proteostasis, and protein trafficking.

**Abstract** How the fate (folding versus degradation) of glycoproteins is determined in the endoplasmic reticulum (ER) is an intriguing question. Monoglucosylated glycoproteins are recognized by lectin chaperones to facilitate their folding, whereas glycoproteins exposing well-trimmed mannoses are subjected to glycoprotein ER-associated degradation (gpERAD); we have elucidated how mannoses are sequentially trimmed by EDEM family members (George et al., 2020; 2021 eLife). Although reglucosylation by UGGT was previously reported to have no effect on substrate degradation, here we directly tested this notion using cells with genetically disrupted UGGT1/2. Strikingly, the results showed that UGGT1 delayed the degradation of misfolded substrates and unstable

glycoproteins including ATF6α. An experiment with a point mutant of UGGT1 indicated that the glucosylation activity of UGGT1 was required for the inhibition of early glycoprotein degradation. These and overexpression-based competition experiments suggested that the fate of glycoproteins is determined by a tug-of-war between structure formation by UGGT1 and degradation by EDEMs. We further demonstrated the physiological importance of UGGT1, since ATF6α cannot function properly without UGGT1. Thus, our work strongly suggests that UGGT1 is a central factor in ER protein quality control via the regulation of both glycoprotein folding and degradation.

## Introduction

The ER plays an essential role in the cell as the site of biosynthesis of approximately one-third of all proteins (*Kaufman, 1999*). Membrane and secretory proteins, after being newly synthesized on ribosomes attached to the ER membrane, are translocated to the ER in a translation-coupled manner (*Brodsky and Skach, 2011*). Post-translational modifications of proteins occurring in the ER include disulfide bond formation mediated by PDI (protein disulfide isomerase) family members and *N*-linked glycosylation of Asn in the consensus sequence consisting of Asn-X-Ser/Thr (X≠Pro). *N*-Glycan modifying a large number of proteins that enter the ER is converted from oligomannose type to complex type in the Golgi and is deeply involved in biological phenomena inside and outside the cell (*Pinho et al., 2023*).

The structure of *N*-glycans plays a key role in protein folding and degradation in the ER (*Berner et al., 2018*; *Ninagawa et al., 2021*). The *N*-glycan added to nascent proteins is composed of three glucose, nine mannoses, and two *N*-acetylglucosamines (GlcNAc), and is termed $Glc_3Man_9GlcNAc_2$ (G3M9), which is processed to GM9 by the actions of glucosidase I and glucosidase II. Calnexin (CNX) and calreticulin (CRT) recognize glycoproteins with GM9 to facilitate productive folding (*Lamriben et al., 2016*) and then the last glucose is removed by glucosidase II. If glycoproteins with M9 attain their tertiary structure, they move on to the next compartment of the secretory pathway (*Ninagawa et al., 2021*). If the protein portion does not form a proper structure, UDP-glucose glycoprotein glucosyltransferses (UGGTs: UGGT1 and UGGT2) re-add glucose to the glycoprotein (reglucosylation) for recognition by CNX/CRT to facilitate protein folding, collectively referred to as the 'CNX/CRT' cycle (*Lamriben et al., 2016*; *Sun and Brodsky, 2019*). UGGT1 promotes substrate solubility (*Ferris et al., 2013*) and prefers proteins with exposed hydrophobic regions to folded proteins (*Caramelo et al., 2004*; *Sousa and Parodi, 1995*). UGGT1 has a paralogue, UGGT2, whose glucosyltransferase activity is weaker than that of UGGT1 at least in humans (*Ito et al., 2020*; *Takeda et al., 2014*).

If glycoproteins with M9 are not folded correctly within a certain period, mannose residues are trimmed from M9 first by EDEM2-S-S-TXNDC11 complex to M8B (*George et al., 2020*; *Ninagawa et al., 2014*) and then by EDEM3/1 to M7A, M6 and M5 exposing α1,6-bonded mannose on the glycoproteins (*George et al., 2021*; *Hirao et al., 2006*; *Ninagawa et al., 2014*), which are recognized by OS9 or XTP3B lectins for degradation (*van der Goot et al., 2018*). They are recruited to the HRD1-SEL1L complex on the ER membrane, retrotranslocated back to the cytosol, and degraded via the ubiquitin-proteasome system (*Figure 1A*). The series of these processes is collectively designated gpERAD (*Ninagawa et al., 2021*; *Smith et al., 2011*).

However, how the fate (folding versus degradation) of glycoprotein is determined in the ER is not clearly understood, and remains one of the biggest issues in the field of ER protein quality control. UGGTs appear to be among the key enzymes determining this. UGGTs have been shown to contribute to glycoprotein folding through their reglucosylation activity (*Helenius and Aebi, 2004*; *Pearse et al., 2010*; *Sousa et al., 1992*; *Tessier et al., 2000*), but seem not to affect ERAD, because it was previously reported that the presence of one glucose in the A branch of *N*-glycans did not change the timing of substrate degradation as follows *Tannous et al., 2015*: MI8-5 Chinese hamster ovary cells are deficient in the dolichol-P-glucose–dependent glycosyltransferase termed Alg6, and therefore produce only glycoproteins with *N*-glycans lacking glucoses (M9). In these cells, *N*-glycans with one glucose are produced only by the action of UGGTs. Accordingly, the monoglucosylated state is maintained by treatment of MI8-5 with the glucosidase inhibitor 1-deoxynojirimycin (DNJ). It was found that such trapping of glycoproteins in a monoglucosylated state delayed their efflux from the ER, as expected, but did not affect their rate of degradation. The selection for the degradation process

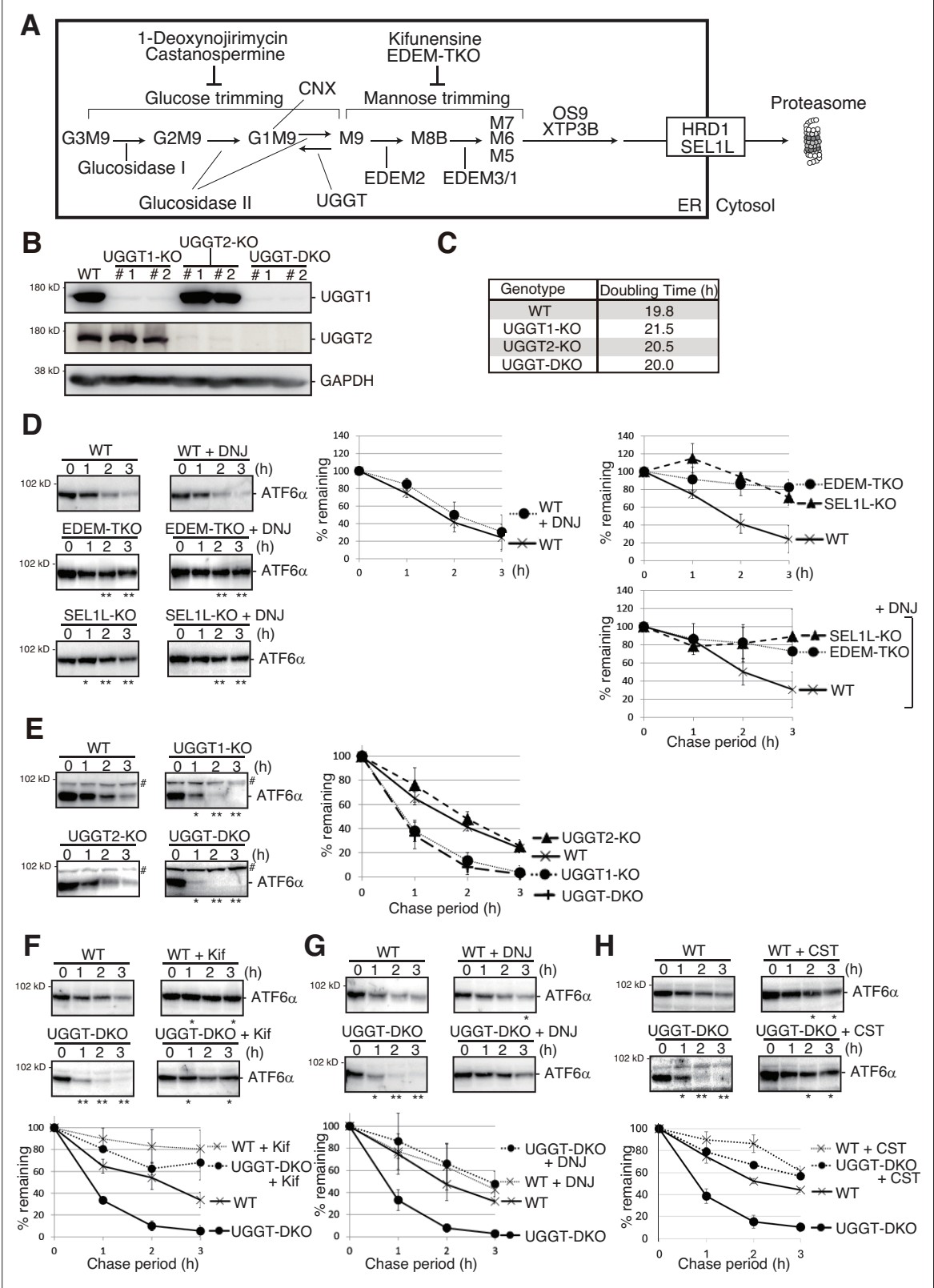

**Figure 1.** Effect of UGGT1/2-KO on degradation of ATF6α. (**A**) Schematic presentation of *N*-glycan processing and glycoprotein ER-associated degradation (gpERAD). (**B**) Immunoblotting to determine endogenous protein expression of UGGT1 and UGGT2 in UGGT1-KO, UGGT2-KO, and UGGT-DKO HCT116 cells using anti-UGGT1, anti-UGGT2, and anti-Glyceraldehyde-3-phosphate dehydrogenase (GAPDH) antibodies. (**C**) Doubling time of wild-type (WT), UGGT1-KO, UGGT2-KO, and UGGT-DKO HCT116 cells (n=3). (**D**) Cycloheximide chase (50 μg/ml) and subsequent

*Figure 1 continued on next page*

*Figure 1 continued*

immunoblotting experiments to determine the degradation rate of endogenous ATF6α in WT, EDEM-TKO and SEL1L-KO HCT116 cells treated with or without 0.5 mM 1-deoxynojirimycin (DNJ) treatment. DNJ was added 2 hr before the addition of CHX. Endogenous ATF6α was detected by immunoblotting using anti-ATF6α antibody. The means from three independent experiments with standard deviations (error bars) are plotted against the chase period (n=3). p-value: *<0.05, **<0.01. (**E**) Cycloheximide chase (50 µg/ml) and subsequent immunoblotting experiments to determine the degradation rate of endogenous ATF6α in WT, UGGT1-KO, UGGT2-KO, and UGGT-DKO HCT116 cells (n=3), as in (**D**) # denotes a non-specific band. p-value: *<0.05, **<0.01. (**F**) Cycloheximide chase (50 µg/ml) and subsequent immunoblotting experiments to determine the degradation rate of endogenous ATF6α in WT and UGGT-DKO HCT116 cells treated with or without 10 µg/ml kifunensine (Kif) (n=3), as in (**D**). Kif was added 1 hr before the addition of CHX. p-value: *<0.05, **<0.01. (**G**) Cycloheximide chase (50 µg/ml) and subsequent immunoblotting experiments to determine the degradation rate of endogenous ATF6α in WT and UGGT-DKO HCT116 cells treated with or without 0.5 mM DNJ (n=3), as in (**D**). DNJ was added 2 hr before the addition of cycloheximide (CHX). p-value: *<0.05, **<0.01. (**H**) Cycloheximide chase (50 µg/ml) and subsequent immunoblotting experiments to determine the degradation rate of endogenous ATF6α in WT and UGGT-DKO HCT116 cells treated with or without 1 mM castanospermine (CST) (n=3), as in (**D**). CST was added 2 hr before the addition of CHX. p-value: *<0.05, **<0.01.

The online version of this article includes the following source data and figure supplement(s) for figure 1:

**Source data 1.** Original files for western blot analysis are displayed in *Figure 1B and D–H*.

**Source data 2.** PDF file containing original membranes of western blots for *Figure 1B and D–H*.

**Figure supplement 1.** Generation of UGGT1/2-KO HCT116 cells.

**Figure supplement 1—source data 1.** Original files for agarose gel analysis are displayed in *Figure 1—figure supplement 1E–H*.

**Figure supplement 1—source data 2.** PDF file containing original agarose gels for *Figure 1—figure supplement 1E–H*.

**Figure supplement 2.** Characterization of UGGT1/2-KO HCT116 cells.

**Figure supplement 2—source data 1.** Original files for western blot and pulse-chase analysis are displayed in *Figure 1—figure supplement 2A–F*.

**Figure supplement 2—source data 2.** PDF file containing original membranes of western blots and gels of pulse-chase analysis for *Figure 1—figure supplement 2A–F*.

**Figure supplement 3.** Generation of SEL1L-KO HCT116 cells.

**Figure supplement 3—source data 1.** Original files for agarose gel and western blot are displayed in *Figure 1—figure supplement 3C–E*.

**Figure supplement 3—source data 2.** PDF file containing original agarose gel and membranes of western blot for *Figure 1—figure supplement 3C–E*.

**Figure supplement 4.** HPLC analysis of *N*-glycan of various types of cells untreated or treated with DNJ.

**Figure supplement 5.** *N*-glycan profiling of various types of cells untreated or treated with 1-deoxynojirimycin (DNJ).

**Figure supplement 6.** Effect of CST on *N*-glycan profiling of cells.

**Figure supplement 6—source data 1.** A original file for western blot are displayed in *Figure 1—figure supplement 6A*.

**Figure supplement 6—source data 2.** PDF file containing an original membrane of western blot for *Figure 1—figure supplement 6A*.

seemed to progress in a dominant manner that was independent of the glucosylation state of the A-branch on the substrate glycans.

We consider that it has not been directly examined whether UGGT-mediated reglucosylation of *N*-glycan is involved in the degradation of glycoproteins in the ER. Here, we generated UGGT1-knockout (KO), UGGT2-KO, and UGGT-double KO (DKO) cell lines to investigate this involvement. Surprisingly, we found that the degradation of misfolded and unstable glycoproteins was markedly accelerated in UGGT1-KO and DKO cells. Our work has identified UGGT1 as a regulator of both protein folding and degradation in the ER and thus emphasizes the importance of UGGT1 in ER protein quality control.

## Results

### Generation of knockout cell lines of UGGTs

To explore the roles of UGGT1 and UGGT2 in ERAD, UGGT1-KO, UGGT2-KO, and UGGT-DKO cell lines were generated from HCT116 diploid cells derived from human colonic carcinoma using Clustered Regularly Interspaced Short Palindromic Repeats (CRISPR)-Cas9. Gene disruptions of UGGT1 and UGGT2 were confirmed at the genome, mRNA, and protein level (*Figure 1B*; *Figure 1—figure supplement 1A–J*). The growth rate of the KO cell population was not significantly altered (*Figure 1C*).

Previously, the protein level of UGGT2 was estimated to be 4% relative to that of UGGT1 in HeLa cells (*Adams et al., 2020*; *Itzhak et al., 2016*). In our hands, the protein expression levels of UGGT2 in

HCT116 and HeLa cells were found to be approximately 4.5% and 26% of that of UGGT1, respectively, as estimated by transfection of UGGT1-Myc3 and UGGT2-Myc3 (*Figure 1—figure supplement 2A*). Immunoblotting showed that ER stress marker proteins BiP, XBP1(S), and ATF4 were not induced in UGGT1-KO, UGGT2-KO, or UGGT-DKO cells (*Figure 1—figure supplement 2B*). Both UGGT1 and UGGT2 are modified with *N*-glycans, as indicated by their sensitivity to treatment with Endoglycosidase H (EndoH), which cleaves oligomannose-type *N*-glycans localized in the ER (*Figure 1—figure supplement 2C*).

In UGGT1-KO and UGGT-DKO cells but not in UGGT2-KO cells, the secretion efficiency of α1-antitrypsin (A1AT) and erythropoietin (EPO) determined by pulse-chase experiments using $^{35}$S was decreased (*Figure 1—figure supplement 2D, E*). In UGGTs-KO cells, maturation of hemagglutinin (HA) from oligomannose type to complex type was delayed, as reported previously (*Figure 1—figure supplement 2F*; *Hung et al., 2022*). These results confirmed the establishment of KO cells for UGGTs, which show the expected phenotype with UGGT1 involved in the maturation of nascent polypeptides.

## Effect of UGGT1/2 knockout on ERAD

We then examined the effect of UGGT1/2 knockout on ERAD. We previously showed that ATF6α functions as an unfolded protein response (UPR) sensor/transducer but is a somewhat unfolded protein that is constitutively subjected to gpERAD (*Haze et al., 1999*; *Horimoto et al., 2013*). The half-life of ATF6α in WT HCT116 cells was less than 2 hr and degradation of ATF6α was blocked almost completely in SEL1L-KO cells (the construction and characterization of these KO cells is described in *Figure 1—figure supplement 3*) and in EDEM1/2/3-triple KO (TKO) HCT116 cells (*Figure 1D*), as was shown previously (*Horimoto et al., 2013*; *Ninagawa et al., 2014*). Strikingly, the degradation of ATF6α was markedly accelerated in UGGT1-KO and UGGT-DKO cells but not in UGGT2-KO cells (*Figure 1E*). These findings revealed for the first time that UGGT1 is involved in ERAD. The fact that ATF6α was stabilized in UGGT-DKO cells treated with a mannosidase inhibitor, kifunensine, as in WT cells treated with kifunensine, indicated that the acceleration of ATF6α degradation still required mannose trimming (*Figure 1F*).

We then examined the effect of the glucosidase inhibitor DNJ, which preferentially inhibits glucosidase II (*Saunier et al., 1982*; *Szumilo et al., 1987*; *Zeng et al., 1997*). Indeed, accumulation of GM9 rather than G3M9 or G2M9 was observed in WT, UGGT-DKO, and EDEM-TKO cells after treatment with DNJ [*Figure 1—figure supplement 4A–C* (raw data), *Figure 1—figure supplement 5A–C* (isomer composition ratio)]. Treatment of WT cells with DNJ did not affect the degradation rate of ATF6α (*Figure 1D*, middle panel), consistent with the previously published claim described in the Introduction. Stabilization of ATF6α in SEL1L-KO and EDEM-TKO cells was not affected by treatment with DNJ (*Figure 1D*, right bottom panel). This is because SEL1L is required for retrotranslocation of substrates, and because mannose trimming required for gpERAD cannot work and, therefore, M9 markedly accumulates in EDEM-TKO cells, compared with WT cells [*Figure 1—figure supplement 4A, C* (raw data) and *Figure 1—figure supplement 5D* (isomer composition ratio)]; note that both retrotranslocation and mannose trimming are events downstream of glucose trimming (*Figure 1A*). Comparison between WT and EDEM-TKO cells indicates that ATF6α is still degraded via gpERAD requiring mannose trimming even in the presence of DNJ (*Figure 1D*, right bottom panel).

Treatment of WT cells with DNJ resulted in a similar degradation rate of ATF6α, however, the accelerated degradation of ATF6α in UGGT-DKO cells was compromised by treatment with DNJ, so that the degradation rate of ATF6α became very similar in WT and UGGT-DKO cells after treatment with DNJ (*Figure 1G*). This implies that the glucose trimming-mediated production rate of M9 is important for efficient gpERAD. Indeed, an increase in the level of GM9 and the decrease in the level of M9 were observed in WT and UGGT-DKO cells treated with DNJ [*Figure 1—figure supplement 4A, B* (raw data) and *Figure 1—figure supplement 5A, B* (isomer composition ratio)]. Nonetheless, we noted that ATF6α was not stabilized completely in WT or UGGT-DKO cells treated with DNJ (*Figure 1G*), unlike in EDEM-TKO and SEL1L-KO cells (*Figure 1D*, right top panel). We consider it likely that this is because mannose trimming still occurred even in the absence of glucose trimming, as evidenced by the increased detection of GM8 in WT and UGGT-DKO cells treated with DNJ [*Figure 1—figure supplement 4A, B* (raw data) *Figure 1—figure supplement 5A, B* (isomer composition rate)], leading to recognition of GM7 by OS9/XTP3B for retrotranslocation (Refer to *Figure 1A*). This notion was supported by the similar results obtained using another inhibitor of glucosidase I

and II, castanospermine (CST) (*Figure 1H*), which was reported to preferentially inhibit glucosidase I (*Saunier et al., 1982*; *Szumilo et al., 1987*; *Zeng et al., 1997*); in WT and UGGT-DKO cells treated with CST, large amounts of G3M9 and G2M9 were indeed detected in addition to GM9, but the increased detection of GM8 should also be noted [*Figure 1—figure supplement 6B, C* (raw data) and *Figure 1—figure supplement 6D* (isomer composition ratio)]. CD3δ-ΔTM-HA, a soluble and low molecular weight gpERAD substrate possessing three *N*-glycans, migrated as a single band in WT cells, as a single band with a slightly higher molecular weight (representing a GM9 form) in WT cells treated with DNJ than that in WT cells, and as multiple higher molecular weight bands (representing a mixture of G3M9, G2M9, and GM9, etc.) in WT cells treated with CST than that in WT cells (*Figure 1—figure supplement 6A*). As the occurrence of mannose trimming in the presence of DNJ (GM9 >GM8>GM7) obscured the effect of reglucosylation on gpERAD in a previous study (*Tannous et al., 2015*), the active role of UGGT1 in gpERAD was demonstrated here for the first time by our gene knockout strategy. We consider that UGGT1-mediated reglucosylation reduces the probability of the mannose trimming that is a prerequisite for gpERAD.

We next examined the effect of UGGT1/2 knockout on the degradation of NHK, a soluble gpERAD substrate. NHK with three *N*-glycosylation sites is severely misfolded because of its C-terminus truncation resulting from TC deletion in the α1-antitrypsin (A1AT) gene (*Sifers et al., 1988*). We found that the degradation of NHK was accelerated in UGGT1-KO and UGGT-DKO cells but not in UGGT2-KO cells (*Figure 2A*), similar to the case of ATF6α. Of note, this acceleration required glucosyltransferase activity, as introduction of myc-tagged WT UGGT1 but not its catalytically inactive mutant, D1358A located in the DxD motif (*Takeda et al., 2014*), into UGGT1-KO cells by transfection decelerated the degradation of NHK (*Figure 2B*). Accordingly, the degradation rate of NHK-QQQ, in which all three *N*-glycosylation sites in NHK were mutated (*Ninagawa et al., 2015*), was not affected by gene disruption of UGGT1 or UGGT2 (*Figure 2C*).

We then examined whether the CNX/CRT cycle competes with gpERAD for substrates. We found that simultaneous overexpression of Myc-CNX and Myc-UGGT1 significantly retarded NHK degradation (*Figure 2D*), while simultaneous overexpression of EDEM2 and EDEM3 accelerated NHK degradation (*Figure 2E*, right top panel), as previously reported (*Hirao et al., 2006*; *Mast et al., 2005*). Importantly, such accelerated degradation of NHK was compromised by co-overexpression of Myc-CNX and Myc-UGGT1 (*Figure 2E*, right bottom panel), revealing a tug of war between the CNX/CRT cycle and gpERAD.

## Effect of UGGT1/2 knockout on folded and functional proteins

We examined the effect of UGGT1/2 knockout on the stability of ER-localized endogenous proteins and found that they were stable both in WT and in UGGT-DKO cells no matter whether they were glycoproteins (Grp170, Sil1, and ribophorin I) or non-glycoproteins (CRT) (*Figure 3A–D* and *Figure 3—figure supplement 1A, B*). Similarly, the activity of the Golgi-resident glycosyltransferase GnT-V (MGAT5) with six putative *N*-glycosylation sites (*Hirata et al., 2023*) was similar in WT, UGGT1-KO, UGGT2-KO and UGGT-DKO cells (*Figure 3E* and *Figure 3—figure supplement 1A*).

Interestingly, a truncated version of rat ribophorin I lacking its C-terminal region (aa333-606), termed rRI332-Flag, (*Figure 3—figure supplement 1A*; *de Virgilio et al., 1998*; *Mueller et al., 2006*) was unstable in WT cells compared with ribophorin I when expressed by transfection, and its degradation rate was accelerated in UGGT1-KO and UGGT-DKO cells but not in UGGT2-KO cells (*Figure 3D*, right panel), similar to ATF6α and NHK.

EMC1 is a type I transmembrane protein with three *N*-glycosylation sites, and is involved in the insertion of membrane proteins into the ER membrane (*Jonikas et al., 2009*; *Shurtleff et al., 2018*). Endogenous EMC1 was a relatively stable protein in both WT and UGGT-DKO cells; however, when EMC1-ΔPQQ lacking the PQQ domain was expressed by transfection, the degradation rate of EMC1-ΔPQQ was accelerated in UGGT1-KO and UGGT-DKO cells but not in UGGT2-KO cells (*Figure 3F*), similar to the case of ribophorin I vs rRI332-Flag. Thus, UGGT1-mediated reglucosylation affects the fate of unstable proteins but not stable proteins.

## UGGT1 is required for the proper functioning of ATF6α

Finally, we investigated the physiological significance of the prevention of early degradation of substrates by UGGT1. Upon ER stress (accumulation of unfolded proteins in the ER), a precursor

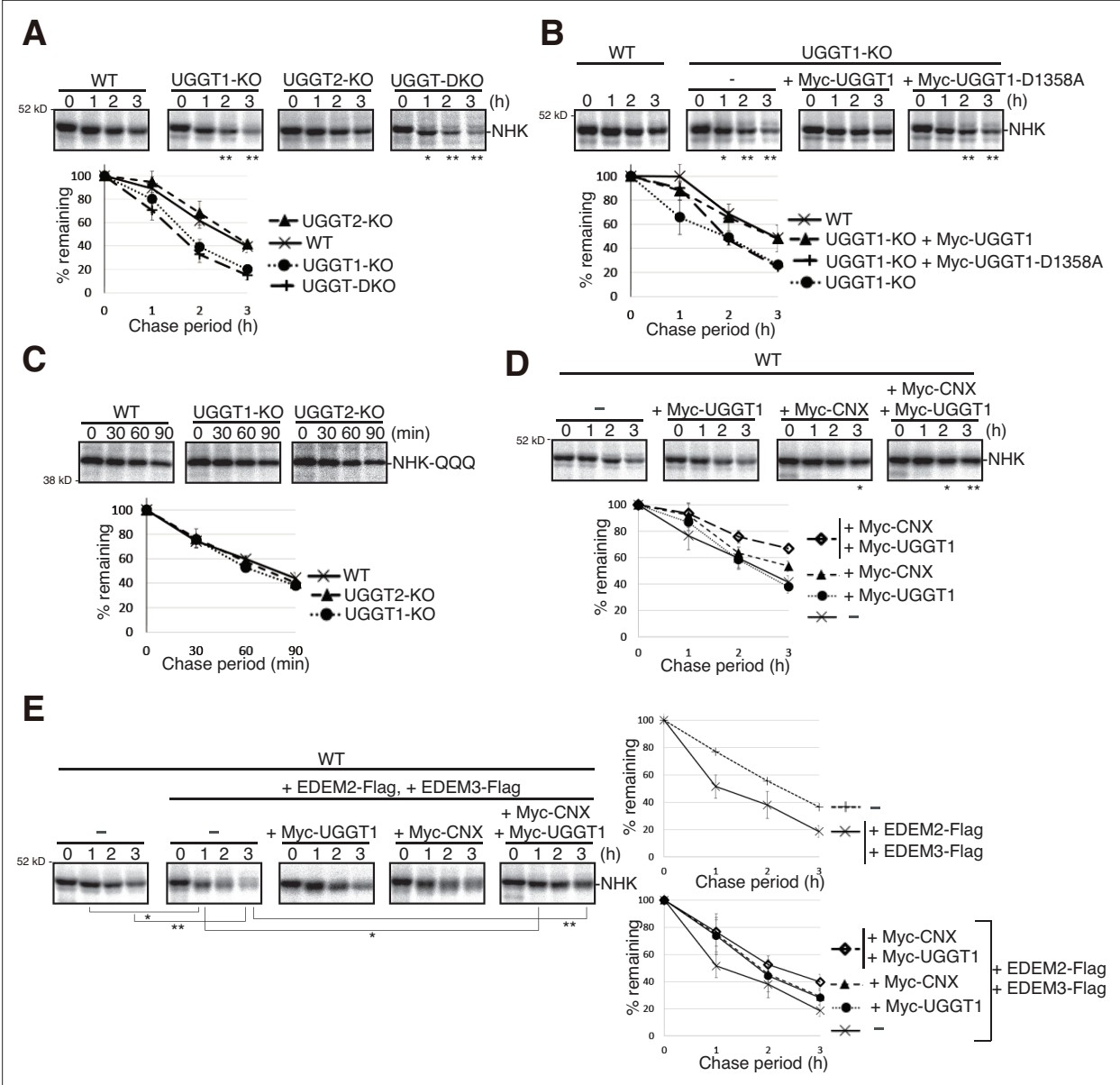

**Figure 2.** Effect of UGGT1/2-KO on degradation of soluble ER-associated degradation (ERAD) substrates. (**A**) Pulse-chase and subsequent immunoprecipitation experiments using anti-α1-PI antibody to determine the degradation rate of null hong kong (NHK) in wild-type (WT), UGGT1-KO, UGGT2-KO, and UGGT-DKO HCT116 cells transfected with plasmid to express NHK (n=3). The radioactivity of each band was determined and normalized with the value at chase period 0 hr. The means from three independent experiments with standard deviations (error bars) are plotted against the chase period (n=3). p-value: *<0.05, **<0.01. (**B**) Rescue experiments using WT and UGGT1-KO HCT116 cells transfected with plasmid to express NHK with or without co-transfected plasmid to express Myc-UGGT1 or Myc-UGGT1-D1358A (n=3), as in (**A**). p-value: *<0.05, **<0.01. (**C**) Pulse-chase and subsequent immunoprecipitation experiments using anti-α1-PI antibody to determine the degradation rate of NHK-QQQ in WT, UGGT1-KO and UGGT2-KO HCT116 cells transfected with plasmid to express NHK-QQQ (n=3), as in (**A**). (**D**) Pulse-chase and subsequent immunoprecipitation experiments using anti-α1-PI antibody to determine the degradation rate of NHK in WT HCT116 cells transfected with plasmid to express NHK with or without co-transfected plasmid to express Myc-UGGT1 and/or Myc-CNX (n=3), as in (**A**). p-value: *<0.05, **<0.01. (**E**) Pulse-chase and subsequent immunoprecipitation experiments using anti-α1-PI antibody to determine the degradation rate of NHK in WT HCT116 cells transfected with plasmid to express NHK together with or without co-transfected plasmid to express EDEM2-Flag, EDEM3-Flag, Myc-UGGT1 and/or Myc-CNX (n=3), as in (**A**). p-value: *<0.05, **<0.01.

The online version of this article includes the following source data for figure 2:

**Source data 1.** Original files for pulse-chase analysis are displayed in *Figure 2A–E*.

**Source data 2.** PDF file containing original gels of pulse-chase for *Figure 2A–E*.

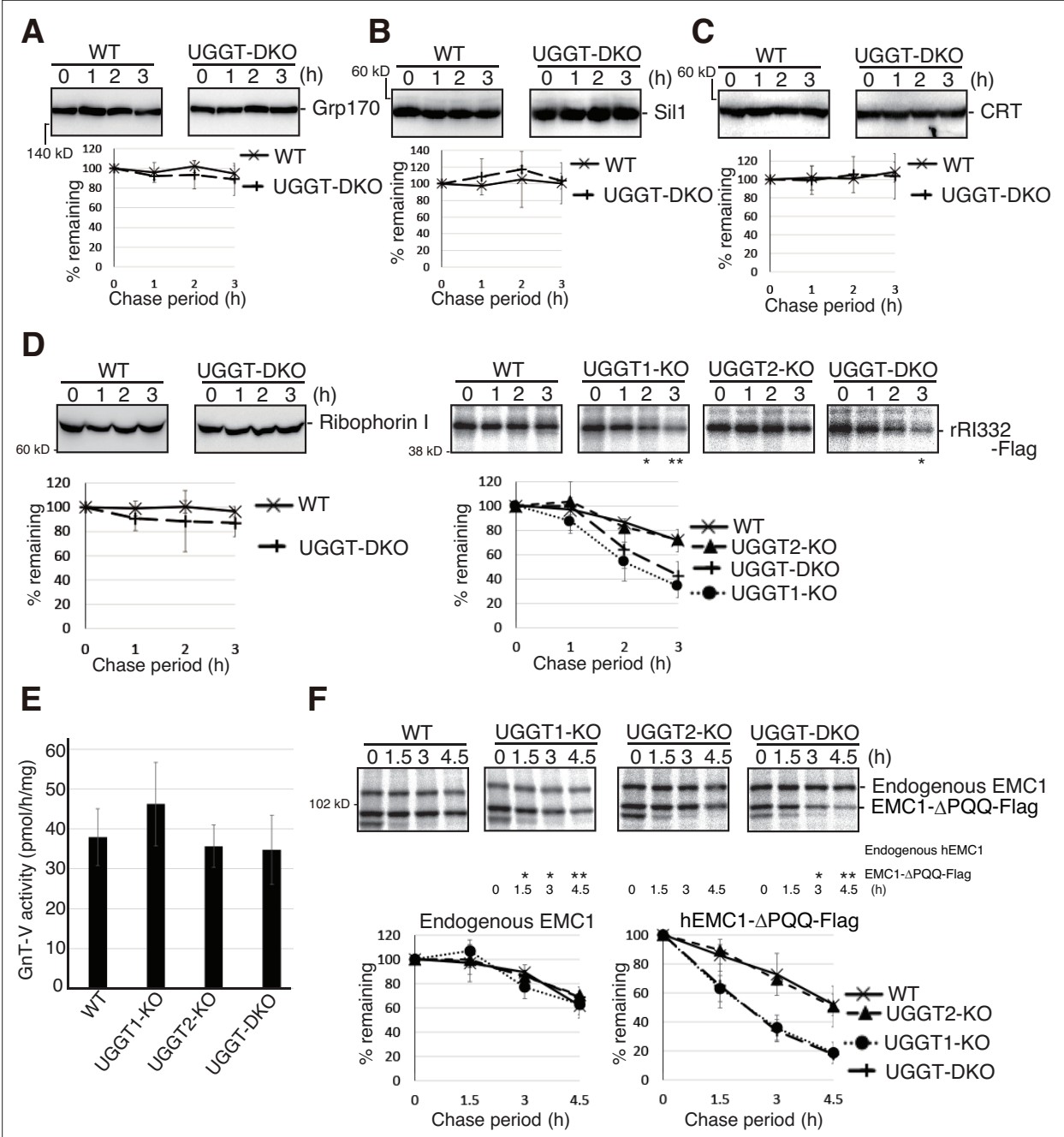

**Figure 3.** Effect of UGGT1/2-KO on functional proteins in the secretory pathway and on their unstable mutants. (**A–C**) Cycloheximide chase (50 µg/ml) and subsequent immunoblotting experiments to determine the degradation rate of endogenous Grp170 (**A**), Sil1 (**B**) and CRT (**C**) in wild-type (WT) and UGGT-DKO HCT116 cells using the respective antibody (n=3). (**D**) [left] Cycloheximide chase and subsequent immunoblotting experiments to determine the degradation rate of endogenous Ribophorin I in WT and UGGT-DKO HCT116 cells using anti-Ribophorin antibody (n=3), as in **Figure 1D**. [right] Pulse-chase and subsequent immunoprecipitation experiments using anti-Flag antibody to determine the degradation rate of rRI332-Flag in WT, UGGT1-KO, UGGT2-KO and UGGT-DKO HCT116 cells transfected with plasmid to express rRI332-Flag (n=3), as in **Figure 2A**. p-value: *<0.05, **<0.01. (**E**) Determination of N-acetylglucosaminyltransferase-V (GnT-V) activity in cell lysates of WT, UGGT1-KO, UGGT2-KO, and UGGT-DKO HCT116 cells (n=3). (**F**) Pulse-chase and subsequent immunoprecipitation experiments using anti-ER membrane protein complex subunit 1 (EMC1) antibody to determine the degradation rate of endogenous EMC1 and EMC1-ΔPQQ-Flag in WT, UGGT1-KO, UGGT2-KO and UGGT-DKO HCT116 cells transfected with plasmid to express EMC1-ΔPQQ-Flag (n=3), as in **Figure 2A**. p-value: *<0.05, **<0.01.

The online version of this article includes the following source data and figure supplement(s) for figure 3:

**Source data 1.** Original files for western blot and pulse-chase analysis are displayed in **Figure 3A–D and F**.

*Figure 3 continued on next page*

*Figure 3 continued*

**Source data 2.** PDF file containing original membranes of western blots and gels of pulse-chase analysis for *Figure 3A–D and F*.

**Figure supplement 1.** Characterization of various proteins present in the secretory pathway.

**Figure supplement 1—source data 1.** Original files for western blot are displayed in *Figure 3—figure supplement 1B*.

**Figure supplement 1—source data 2.** PDF file containing membranes of a western blot for *Figure 3—figure supplement 1B*.

form of ATF6α,designated ATF6α(P), is cleaved at the Golgi to produce the nucleus-localizing form of ATF6α, designated ATF6α(N), which transcriptionally activates promoters containing cis-acting ER stress response element (ERSE) or UPR element (UPRE) (*Mori, 2000*; *Yoshida et al., 1998*), but not the ATF4 promoter, which is translationally activated in response to ER stress (*Lu et al., 2004*). Degradation of ATF6α(P) was accelerated in UGGT1-KO and UGGT-DKO cells but not in UGGT2-KO cells (*Figure 1E*), and accordingly, the protein expression level of endogenous ATF6α(P) was significantly decreased in UGGT1-KO and UGGT-DKO cells but not in UGGT2-KO cells (*Figure 4A*). We noticed conformational change in ATF6α(P), as the remaining ATF6α(P) in UGGT1-KO cells tended to have a more rigid structure that averts degradation, as suggested by its slightly lower sensitivity to trypsin (*Figure 4B*); a trypsin digestion assay can evaluate the rigidity of the protein structure (*George et al., 2020*; *Liu et al., 2016*; *Ninagawa et al., 2015*).

The conversion from ATF6α(P) to ATF6α(N) is a hallmark of ATF6α activation. The amount of ATF6α(N) was significantly decreased in UGGT1-KO and UGGT-DKO cells but not in UGGT2-KO cells compared with WT cells after the treatment with thapsigargin (Tg), an inhibitor of the ER calcium pump, which thereby induces ER stress (*Figure 4C–E*).

Overall transcriptional activity of ATF6α was determined using the ERSE, UPRE, and ATF4 reporters. In WT cells, the ERSE and UPRE reporters were induced 4.8 and 12.1-fold by Tg treatment, respectively, and such induction was abolished in ATF6α-KO cells, as expected, whereas the ATF4 reporter was induced similarly in WT and ATF6α-KO cells (*Figure 4F–H*). In UGGT1-KO and UGGT-DKO cells, the induction level of the ERSE and UPRE reporters was significantly decreased, whereas that of the ATF4 reporter was rather increased in response to Tg treatment (*Figure 4F–H*). Accordingly, UGGT1-KO and UGGT-DKO cells exhibited sensitivity to Tg treatment compared with WT cells (*Figure 4I*) (see DISCUSSION for responses in UGGT2-KO cells). These results show that UGGT1 is required for the proper functioning of ATF6α.

## Discussion

Our results reveal that UGGT1 contributes to the determination of the fate of glycoproteins in the ER. In WT cells, unfolded or misfolded glycoproteins with M9 are subjected to either the CNX/CRT cycle for folding via UGGT1-mediated reglucosylsation or gpERAD via EDEMs-mediated mannose trimming, leading to degradation at a certain rate [the half-life of ATF6α was ~1.5 hr (*Figure 1E*) and that of NHK was ~2.5 hr (*Figure 2A*)]. In UGGT1-KO cells, unfolded or misfolded glycoproteins with M9 skip the CNX/CRT cycle and are subjected to gpERAD, resulting in accelerated degradation [the half-life of ATF6α became less than 1 h (*Figure 1E*) and that of NHK became ~1.5 hr (*Figure 2A*)]. This acceleration occurred for rRI332 (*Figure 3D*, right panel) and hEMC1-ΔPQQ (*Figure 3F*), but not for native and stable glycoproteins, such as Ribophorin I, EMC1, Grp170, and Sil1 (*Figure 3A, B, D, and F*). Glycoproteins with stable conformational states do not require UGGT1, whereas unstable, unassembled, conformationally abnormal, or severely misfolded proteins are examined by UGGT1 for the possibility of proper folding in the CNX/CRT cycle. Therefore, we conclude that the capability of reglucosylation has an impact on determining the timing of substrate degradation, and UGGT1 acts to prevent early degradation. Substrate degradation was delayed by knockout of EDEMs (*George et al., 2020*; *Leto et al., 2019*; *Ninagawa et al., 2015*; *Ninagawa et al., 2014*) and accelerated by overexpression of EDEMs (*Figure 2E*; *Hirao et al., 2006*; *Mast et al., 2005*). In contrast, substrate degradation was accelerated by knockout of UGGT1 (*Figures 1E and 2A* and *Figure 3D–F*) and was delayed by overexpression of CNX and UGGT1 (*Figure 2D and E*). It is further concluded that UGGT1 plays a 'tug-of-war' with EDEMs regarding the determination of the fate of glycoproteins: whether to give them a chance to fold, or to degrade them (*Figure 5A*).

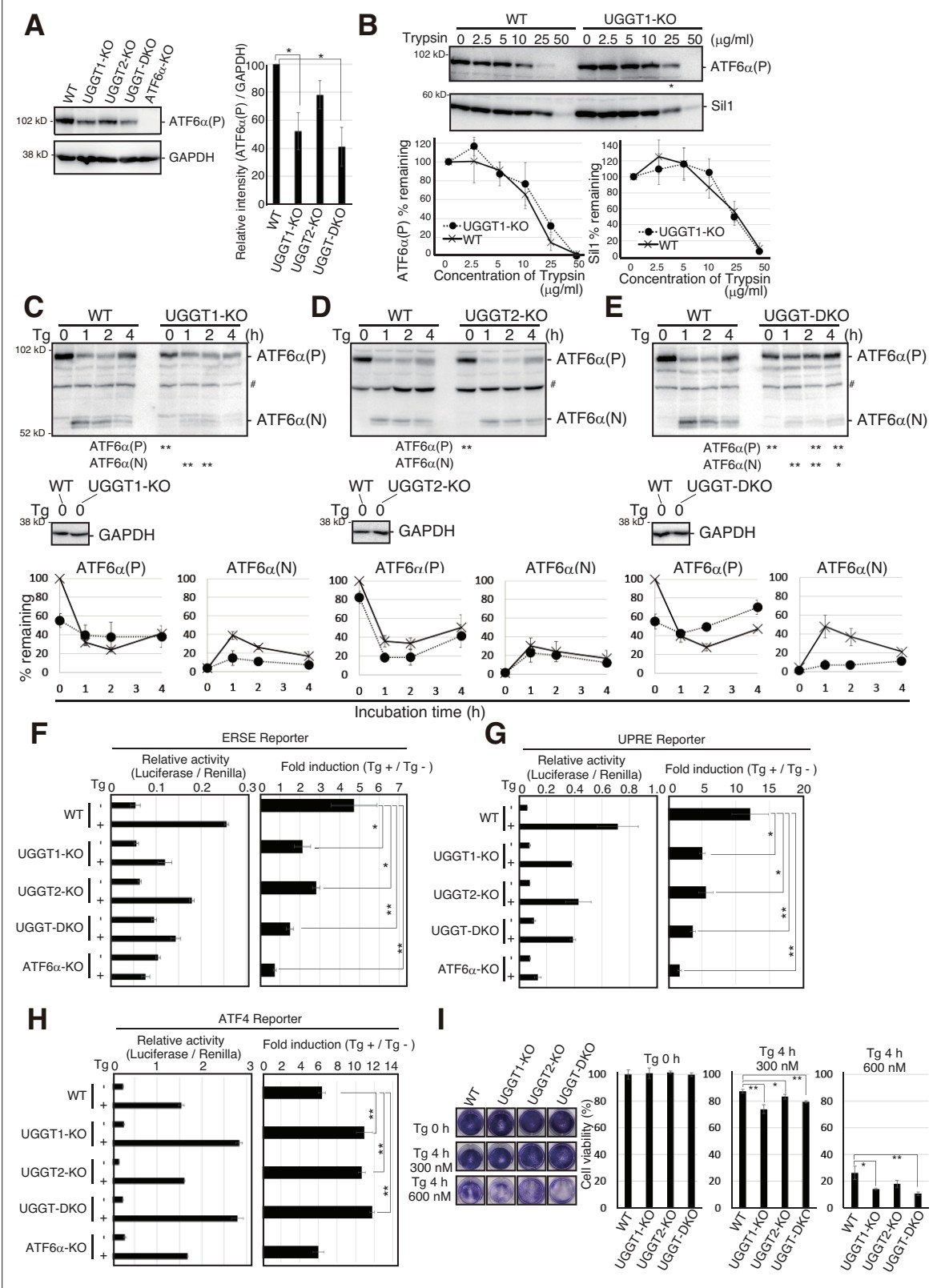

**Figure 4.** Effects of UGGT1-KO on ATF6α-mediated unfolded protein response (UPR). (**A**) Immunoblotting to determine the protein level of endogenous ATF6α(P) in wild-type (WT), UGGT1-KO, UGGT2-KO, UGGT-DKO, and ATF6α-KO HCT116 cells using anti-ATF6α antibody (n=3). p-value: *<0.05. (**B**) The same amounts of proteins in cell lysates from WT and UGGT1-KO cells were treated with the indicated concentration of trypsin for 15 min at room temperature. Enzymatic activity of trypsin was terminated by the addition of Laemmli SDS sample buffer containing 100 mM DTT and

*Figure 4 continued on next page*

*Figure 4 continued*

a 10 x protease inhibitor cocktail. Immunoblotting was conducted as in *Figure 1D*, using anti-ATF6α and anti-Sil1 antibodies. In electrophoresis for the detection of ATF6α, loaded UGGT1-KO samples contained twice as much protein as loaded WT samples to compensate for the decrease in the level of ATF6α(P). (**C–E**) WT, UGGT1-KO, UGGT2-KO, and UGGT-DKO HCT116 cells were incubated in the presence of 300 nM thapsigargin (Tg) for the indicated periods. The levels of endogenous ATF6α(P), ATF6α(N), and Glyceraldehyde-3-phosphate dehydrogenase (GAPDH) were determined by immunoblotting. Intensity of the unstressed ATF6α(P) band (0 hr) was set to 100%. # indicates a non-specific band. p-value: *<0.05, **<0.01. (**F–H**) WT, UGGT1-KO, UGGT2-KO, UGGT-DKO, and ATF6α-KO HCT116 cells were transiently transfected with the ER stress responsive element (ERSE) (**D**), UPR element (UPRE) (**E**), or activating transcription factor 4 (ATF4) (**F**) reporters. Twenty-four hours after the transfection, cells were treated with 300 nM thapsigargin (Tg) for 6 hr and then harvested to determine luciferase activity (n=3). Relative activity of Luciferase to Renilla, and fold induction of Tg + relative to Tg – are shown. p-value: *<0.05, **<0.01. (**I**) Crystal violet assay to determine endoplasmic reticulum (ER) stress sensitivity of WT, UGGT1-KO, UGGT2-KO, and UGGT-DKO HCT116 cells. An equal number of cells of each cell line treated with 300 nM or 600 nM Tg for 4 hr were spread on 24-well plates and cultured without Tg for 5 d. Resulting HCT116 cells were stained with crystal violet and photographed, and then stained cells were solubilized with 1% SDS, and absorbance of the resulting solution was measured at 570 nm. Cell viability in ER-stressed cells was calculated as the ratio of A570 relative to that obtained with unstressed cells. p-value: *<0.05, **<0.01.

The online version of this article includes the following source data for figure 4:

**Source data 1.** Original files for western blot analysis are displayed in *Figure 4A–E*.

**Source data 2.** PDF file containing original membranes of western blots for *Figure 4A–E*.

Previously, in higher animals, the reglucosylation activity of UGGT1 was thought to contribute to folding and secretion but not to the degradation of glycoproteins, based on experiments using the glucosidase inhibitor DNJ (*Tannous et al., 2015*). The reason why the degradation rate was not changed significantly by DNJ treatment can be explained by the occurrence of mannose trimming in the presence of DNJ: GM8 was increasingly detected in cells treated with DNJ (*Figure 1—figure supplement 5A–C*). Substrates with GM9 are degraded via mannose trimming (GM9 >GM8>GM7) in cells treated with DNJ. Nonetheless, as ATF6α was degraded at a similar rate in WT cells, WT cells treated with DNJ, and UGGT1-KO cells treated with DNJ (*Figure 1G*), we consider it likely that mannose trimming is inefficient when a glucose residue is present on the A chain (*Figure 5*).

Although it must be taken into account that the protein expression level of UGGT2 was markedly lower than that of UGGT1 in HCT116 compared to HeLa cells, the inhibitory effect of UGGT2 on substrate degradation appears to be quite limited in HCT116 cells. For all substrates used in this study, genetic disruption of UGGT2 alone did not accelerate degradation (*Figures 1E and 2A* and *Figure 3D and F*), and overexpression of UGGT2 in UGGT-DKO cells did not attenuate the accelerated degradation of NHK, whereas overexpression of UGGT1 did so (data not shown). Instead, we found that the loss of UGGT2 mitigated the transcriptional activity of ATF6α (*Figure 4F and G*) without affecting the ER stress-induced cleavage of ATF6α (*Figure 4D*). UGGT2 may have other yet unknown direct or indirect effects on ATF6α. Since UGGT2 has been shown to decrease saturated fatty acids through lipid glucosylation (*Hung et al., 2022*), lipid status may have some effect on the activity of ATF6α.

Our findings strongly suggest that UGGT1 is not just a CNX/CRT safety net, but should be considered to be a central component involved in the *N*-glycan-dependent folding and degradation in the ER. The loss of UGGT1 reduced tolerance to ER stress in cultured cells (*Figure 4I*). UGGT1 is necessary for the proper functioning of ER-resident proteins such as ATF6α (*Figure 4C–H*). It is highly possible that ATF6α undergoes structural maintenance by UGGT1, which would avoid unnecessary degradation and maintain proper function, because ATF6α with a more rigid structure tended to remain in UGGT1-KO cells (*Figure 4B*). Responses of ERSE- and UPRE-containing promoters toward ER stress, which require ATF6α, were decreased in UGGT1-KO cells (*Figure 4F and G*). In contrast, ATF4 reporter activity was increased in UGGT1-KO cells (*Figure 4H*). Given that the basal expression level of ATF4 in UGGT1-KO cells was comparable to that in WT (*Figure 1—figure supplement 2B*), the ATF4 pathway might be activated to partially compensate for the functional loss of the ERSE and UPRE pathway in UGGT1-KO cells in response to acute ER stress.

Our work focusing on the function of mammalian UGGT1 greatly advances the understanding of how ER homeostasis is maintained in higher animals. Considering that *Saccharomyces cerevisiae* does not have a functional orthologue of UGGT1 (*Ninagawa et al., 2021*) and that KO of UGGT1 causes embryonic lethality in mice (*Molinari et al., 2005*), it would be interesting to know at what point the function of UGGT1 became evolutionarily necessary for life. Related to its importance in animals,

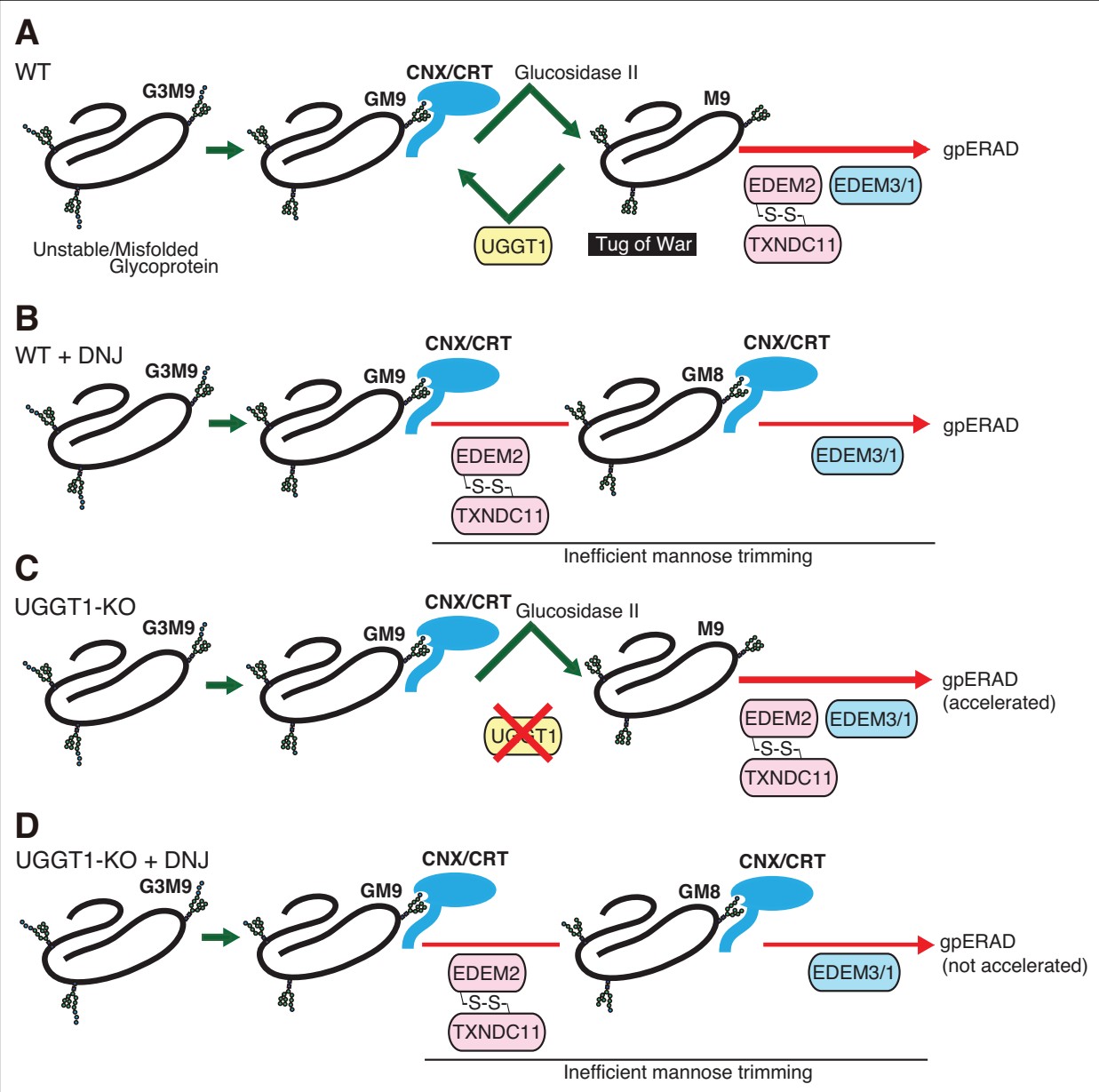

**Figure 5.** Proposed fate of glycoproteins in the endoplasmic reticulum (ER). (**A**) In wild-type (WT) cells, glucosidase II produces glycoproteins with M9, which are either reglucosylated by UGGT1 and then subjected to the calnexin (CNX)/calreticulin (CRT) cycle, or degraded via EDEMs-mediated mannose trimming. Thus, the fate of glycoproteins is determined by a tug-of-war between structure formation by UGGT1 and degradation by EDEMs. (**B**) In WT cells treated with 1-deoxynojirimycin (DNJ), glycoproteins are not subjected to the CNX/CRT cycle and are degraded by EDEMs-mediated glycoprotein ER-associated degradation (gpERAD). It is likely that mannose trimming from GM9 proceeds more slowly than that from M9, so that glycoproteins are degraded at a similar speed in (**A**) and (**B**). (**C**) In UGGT1-knockout (KO) cells, glycoproteins are directly subjected to premature (accelerated) degradation by EDEMs-mediated gpERAD. (**D**). In UGGT1-KO cells treated with DNJ, glycoproteins are degraded similarly to (**B**).

it would also be of interest to know what kind of diseases UGGT1 is associated with. Recently, it has been reported that UGGT1 is involved in the ER retention of Trop-2 mutant proteins, which are encoded by a causative gene of gelatinous drop-like corneal dystrophy (*Tax et al., 2024*). Not only this, but since the ER is known to be involved in over 60 diseases (*Guerriero and Brodsky, 2012*), we must investigate how UGGT1 and other ER molecules are involved in diseases.

# Materials and methods

**Key resources table**

| Reagent type (species) or resource | Designation | Source or reference | Identifiers | Additional information |
|---|---|---|---|---|
| Cell line | Colorectal carcinoma | ATCC | HCT116; RRID:CVCL_0291 | Authenticated |
| Recombinant DNA reagent | p3xFlag-CMV-14 | *Ninagawa et al., 2014* | | |
| Recombinant DNA reagent | DT-A-pA/loxP/PGK-Puro-pA/loxP | *Ninagawa et al., 2014* | | |
| Recombinant DNA reagent | DT-A-pA/loxP/PGK-Hygro-pA/loxP | *Tsuda et al., 2019* | | |
| Antibody | anti-HA; Rabbit polyclonal | Recenttec | Cat#:R4-TP1411100 | WB (1:1000) |
| Antibody | anti-HA; Mouse polyclonal | Recenttec | Cat#:R4-TM1422100 | WB (1:1000) |
| Antibody | anti-UGGT1; Rabbit polyclonal | Sigma | Cat#:HPA015127 | WB (1:1000) |
| Antibody | anti-UGGT2; Rabbit polyclonal | GeneTex | Cat#:GTX103837; RRID:AB_11164993 | WB (1:1000) |
| Antibody | anti-GAPDH; Rabbit polyclonal | Trevigen | Cat#:2275-PC-100 | WB (1:1000) |
| Antibody | anti-Myc; Rabbit polyclonal | MBL | Cat#:MBL562 | WB (1:1000) |
| Antibody | anti-Myc; Mouse monoclonal | Wako | Cat#:011–21874 | WB (1:1000) |
| Antibody | anti-ATF6; Rabbit polyclonal | *Haze et al., 1999* | | WB (1:1000) |
| Antibody | anti-A1AT; Rabbit polyclonal | Dako | Cat#:A0012 | WB (1:1000) |
| Antibody | anti-Flag; Mouse monoclonal | Sigma | Cat#:F3165 | WB (1:1000) |
| Antibody | anti-Grp170; Rabbit polyclonal | GeneTex | Cat#:GTX102255; RRID:AB_1950534 | WB (1:1000) |
| Antibody | anti-Sil1; Rabbit polyclonal | GeneTex | Cat#:GTX116755; RRID:AB_10617803 | WB (1:1000) |
| Antibody | anti-Ribophorin I; Rabbit monoclonal | Abcam | Cat#:ab198508 | WB (1:1000) |
| Antibody | anti-CRT; Rabbit monoclonal | Enzo Life Sciences | Cat#:ADI-SPA-600; RRID:AB_10618853 | WB (1:1000) |

## Statistics

Statistical analysis was conducted using Student's t-test, with probability expressed as \*p<0.05 and \*\*p<0.01 for all figures.

## Construction of plasmids

Recombinant DNA techniques were performed using standard procedures (*Sambrook et al., 1989*) and the integrity of all constructed plasmids was confirmed by extensive sequencing analyses. Using 3xMyc-Fw and 3xMyc-Rv primers, a Myc3 fragment was obtained from ATF6α(C)-TAP2 (Myc3-TEV-ProteinA) (*George et al., 2021*) and inserted into A p3xFlag-CMV-14 expression vector (Sigma) at the site of BamHI to construct p3xMyc-CMV-14 expression vector. Full-length open reading frame of human UGGT1 or UGGT2 was amplified using PrimeSTAR HS DNA polymerase and a pair of primers, namely, UGGT1-cloningFw and UGGT1-cloningRv for UGGT1, and UGGT2-cloningFw and

UGGT2-cloningRv for UGGT2, respectively, from a cDNA library of HCT116 which was prepared using Moloney murine leukemia virus reverse transcriptase (Invitrogen), as described previously (*Ninagawa et al., 2014*). Site-directed mutagenesis was carried out with DpnI to construct UGGT1-D1358A-Myc3 using UGGT1-D1358AFw and UGGT1-D1358Arv primers and DpnI. A partial open reading frame of Rat Ribophorin I was amplified using a pair of primers, namely, RatRI332-cloningFw and RatRI332-cloningRv, and inserted into p3xFlag-CMV-14 or p3xMyc-CMV-14 between the HindIII and KpnI sites to construct RI332-Flag or RI332-Myc, respectively. Expression vectors of NHK, CD3-δ-ΔTM-HA, CD3-δ-ΔTM-<33–7aa-34>-HA, EMC1-ΔPQQ-Flag, A1AT, and Hemagglutinin were described previously (*Ninagawa et al., 2015*; *Ninagawa et al., 2014*; *Ninagawa et al., 2020*).

## Cell culture and transfection

HCT116 cells (ATCC CCL-247) and HeLa cells (ATCC CCL-2) were cultured in Dulbecco's modified Eagle's medium (glucose 4.5 g/liter) supplemented with 10% fetal bovine serum, 2 mM glutamine, and antibiotics (100 U/ml penicillin and 100 mg/ml streptomycin) at 37 °C in a humidified 5% $CO_2$/95% air atmosphere. Transfection was performed using polyethylenimine max (Polyscience) according to the manufacturer's instructions. EndoH was obtained from Calbiochem; cycloheximide from Sigma; MG132 from Peptide Institute; and Z-vad-fmk from Promega. Absence of mycoplasma contamination was confirmed using MycoBlue Mycoplasma Detector (Vazyme). STR profiling was conducted.

## Immunological techniques

Immunoblotting analysis was carried out according to the standard procedure (*Sambrook et al., 1989*) as described previously (*Ninagawa et al., 2011*). Chemiluminescence obtained using Western Blotting Luminol Reagent (Santa Cruz Biotechnology) was detected using a LAS-3000 mini Lumino-Image analyzer (Fuji Film). The antibodies used are listed in the Supplemental material. Anti-human ATF6α (*Haze et al., 1999*) and EMC1 (*Ninagawa et al., 2015*) antibodies were produced previously. Immunoprecipitation was performed using the described antibodies and protein G- or A-coupled Sepharose beads (GE Healthcare). Beads were washed with high salt buffer (50 mM Tris/Cl, pH 8.0, containing 1% NP-40 and 150 mM NaCl) twice, washed with PBS, and boiled in Laemmli's sample buffer.

## *N*-glycan profiling

Pyridylamination and structural identification of *N*-glycans of total cellular glycoproteins were performed as described previously (*Horimoto et al., 2013*; *Ninagawa et al., 2014*).

## Pulse-chase experiments

Pulse-chase experiments using 9.8 Mbq per dish of EASY-TAG EXPRESS Protein labeling mix [35S] (PerkinElmer) and subsequent immunoprecipitation using suitable antibodies and protein G or A-coupled Sepharose beads (GE Healthcare) were performed according to our published procedure (*Ninagawa et al., 2014*).

## CRISPR/Cas9 method to generate KO cell lines of UGGT1

Using the pair of primers UGGT1sgRNAFw and Rv, the sequence of the BbsI site of px330 (Addgene) was converted to that to express sgRNA for cleavage at exon 2 of the UGGT1 gene. PuroR and backbone fragments were amplified by PCR from DT-A-pA-loxP-PGK-Puro-pA-loxP (*Ninagawa et al., 2014*) using UGGT1-PuroFw and Rv primers, and UGGT1-BackboneFw and UGGT1-BackboneRv primers, respectively. Left and right arms were amplified by PCR from the human genome originated from HCT116 using UGGT1-LarmFw and Rv, and UGGT1-RarmFw and Rv. Four fragments were built up using an NEBuilder HiFi DNA Assembly Cloning Kit (New England Biolabs) to create pKO-hUGGT1-Puro, which was transfected into HCT116 cells with sgRNA expression vector for UGGT1. Clones with puromycin (0.5 μg/ml) resistance were selected.

## CRISPR/Cas9 method to generate KO cell lines of UGGT2

Using the pair of primers UGGT2sgRNAFw and Rv, the sequence of the BbsI site of px330 (Addgene) was converted to that to express sgRNA for cleavage at exon 4 of the UGGT2 gene. Hygro$^r$ and backbone fragments were amplified by PCR from DT-A-pA-loxP-PGK-Hygro-pA-loxP (*Tsuda et al., 2019*)

using UGGT2-HygroFw and Rv primers, and UGGT2-BackboneFw and UGGT2-BackboneRv primers, respectively. Left and right arms were amplified by PCR from the human genome originated from HCT116 using UGGT2-LarmFw and Rv primers, and UGGT2-RarmFw and Rv primers, respectively. Four fragments were built up using an NEBuilder HiFi DNA Assembly Cloning Kit to create pKO-hUGGT2-Hygro, which was transfected into HCT116 cells with sgRNA expression vector for UGGT2. Clones with hygromycin (300 µg/ml) resistance were selected.

## TALEN method to generate KO cell lines of SEL1L

Platinum TALEN plasmid was constructed as described previously (*Ninagawa et al., 2014*; *Sakuma et al., 2013*). In brief, each DNA-binding module was assembled into ptCMV-153/47-VR vectors using the two-step Golden Gate cloning method. The assembled sequence was 5-TGCTGCTGTGTGCGGT GCTgctgagcttggccTCGGCGTCCTCGGGTCA-3, where uppercase and lowercase letters indicate TALEN target sequences and spacer sequences, respectively.

## Reporter assay

Twenty-four hours after transfection, HCT116 cells cultured in a 24-well plate were washed with PBS and lysed in Luciferase Assay Lysis Buffer (Toyo Bnet). Luciferase activity was determined using Pica-Gene Dual-luciferase reporter assay reagent (Toyo Bnet). Relative luciferase activity was defined as the ratio of firefly luciferase activity to renilla luciferase activity. ERSE, UPRE, and ATF4 reporters were described previously (*Saito et al., 2022*). Briefly, the ERSE reporter is pGL3-GRP78(–132)-Luc carrying the human BiP promoter, the UPRE reporter carries p5xUPRE-GL3 identical to p5xATF6GL3, and the ATF4 reporter carries the promoter region of murine ATF4 from position –261 to +124 (ORF starts at +1).

## Measurement of glycosyltransferase activities

The experiment was described previously (*Hirata et al., 2023*). Briefly, 3 µl of the cell lysates was incubated in a total of 10 µl of a reaction buffer [125 mM MES (pH 6.25), 10 mM $MnCl_2$, 200 mM GlcNAc, 0.5% Triton X-100, and 1 mg/ml BSA] supplemented with 20 mM UDP-GlcNAc and 10 µM fluorescence-labeled biantennary acceptor $N$-glycan substrate GnGnbi-PA (PA, 2-aminopyridine) at 37 °C for 3 hr. After the reaction, the sample was boiled at 99 °C for 2 min to inactivate the enzymes and then 40 µl of water was added. After centrifugation at 21,500 $g$ for 5 min, the supernatants were analyzed by reverse-phase HPLC with an ODS column (4.6×150 mm, TSK gel ODS-80TM; TOSOH Bioscience). HPLC analysis was conducted in the isocratic mode in which 80% buffer A (20 mM ammonium acetic buffer [pH 4.0]) and 20% buffer B (1% butanol in buffer A) were loaded at 1 ml/min.

## Trypsin digestion assay

The trypsin digestion assay was described previously (*Ninagawa and Mori, 2016*; *Ninagawa et al., 2015*).

## Crystal violet assay

The crystal violet assay was described previously (*Yamamoto et al., 2007*). Cells spread in 24-well dishes were photographed.

## Acknowledgements

The authors declare no competing financial interests. We are grateful to the members of Biosignal Research Center, Graduate School of Agriculture, Kobe University, and Graduate School of Science, Kyoto University for helpful discussions and encouragement. We appreciate the kind cooperation of the Research Facility Center for Science and Technology, Kobe University. We thank Ms. Shino Oguri, Ms. Kaoru Miyagawa, Ms. Makiko Sawada and Ms. Yuko Tokoro for their technical and secretarial assistance, and Dr. Masayuki Yokoi (Kobe University) for his help in the use of GloMax. This work was financially supported in part by the Ministry of Education, Culture, Sports, Science and Technology, MEXT, Japan (18K06216 to SN, 21H02625 and 23H03838 to HY, 24H00599 to KK, and 17H01432, 17H06419 and 22H00407 to KM), JST-CREST (JP MJCR21E3 to KK), the Takeda Science Foundation (to SN), the Kobayashi Foundation (to SN), the Naito Foundation (to SN), Nagase Science and Technology Foundation (to SN), and JST FOREST Program (grant Number JPMJFR2255 to HY), and

a donation from Dr. Takahiko Nagamine of Sunlight Brain Research Center (to SN). This work was supported by Joint Research of the Exploratory Research Center on Life and Living Systems (ExCELLS) (ExCELLS program No, 21–307) and the Assisted Joint Research Program (Exploration Type) of the J-GlycoNet cooperative network (Support-18), which is accredited by MEXT, Japan, as a Joint Usage/ Research Center.

## Additional information

### Funding

| Funder | Grant reference number | Author |
|---|---|---|
| Japan Society for the Promotion of Science | 18K06216 | Satoshi Ninagawa |
| Japan Society for the Promotion of Science | 21H02625 | Hirokazu Yagi |
| Japan Society for the Promotion of Science | 23H03838 | Hirokazu Yagi |
| Japan Society for the Promotion of Science | 24H00599 | Koichi Kato |
| Japan Society for the Promotion of Science | 17H01432 | Kazutoshi Mori |
| Japan Society for the Promotion of Science | 17H06419 | Kazutoshi Mori |
| Japan Society for the Promotion of Science | 22H00407 | Kazutoshi Mori |
| Japan Science and Technology Agency | 10.52926/JPMJCR21E3 | Kazutoshi Mori |
| Takeda Medical Research Foundation | | Satoshi Ninagawa |
| Kobayashi Foundation | | Satoshi Ninagawa |
| Naito Foundation | | Satoshi Ninagawa |
| Nagase Science Technology Foundation | | Satoshi Ninagawa |
| Japan Science and Technology Agency | JPMJFR2255 | Hirokazu Yagi |
| Sunlight Brain Research Center | | Satoshi Ninagawa |

The funders had no role in study design, data collection and interpretation, or the decision to submit the work for publication.

### Author contributions

Satoshi Ninagawa, Conceptualization, Resources, Data curation, Supervision, Funding acquisition, Validation, Investigation, Writing – original draft, Project administration, Writing – review and editing; Masaki Matsuo, Deng Ying, Shuichiro Oshita, Yasuhiko Kizuka, Data curation, Investigation; Shinya Aso, Kazutoshi Matsushita, Mai Taniguchi, Akane Fueki, Moe Yamashiro, Investigation; Kaoru Sugasawa, Tetsushi Sakuma, Takashi Yamamoto, Resources; Shunsuke Saito, Koshi Imami, Resources, Investigation; Hirokazu Yagi, Koichi Kato, Data curation, Investigation, Methodology; Kazutoshi Mori, Conceptualization, Resources, Supervision, Writing – original draft, Project administration, Writing – review and editing

### Author ORCIDs

Satoshi Ninagawa (iD) https://orcid.org/0000-0002-8005-4716
Masaki Matsuo (iD) https://orcid.org/0009-0004-6540-3238

Shuichiro Oshita [ID] https://orcid.org/0009-0009-5815-0902
Shinya Aso [ID] https://orcid.org/0009-0001-7242-5193
Mai Taniguchi [ID] https://orcid.org/0009-0002-0140-8406
Akane Fueki [ID] https://orcid.org/0009-0003-8091-5103
Moe Yamashiro [ID] https://orcid.org/0009-0006-8723-5436
Kaoru Sugasawa [ID] https://orcid.org/0000-0001-7937-4053
Shunsuke Saito [ID] https://orcid.org/0000-0002-1331-811X
Koshi Imami [ID] https://orcid.org/0000-0002-7451-4982
Yasuhiko Kizuka [ID] https://orcid.org/0000-0002-3181-9743
Tetsushi Sakuma [ID] https://orcid.org/0000-0003-0396-1563
Hirokazu Yagi [ID] https://orcid.org/0000-0001-9296-0225
Koichi Kato [ID] https://orcid.org/0000-0001-7187-9612
Kazutoshi Mori [ID] https://orcid.org/0000-0001-7378-4019

Reviewer #1 (Public review): https://doi.org/10.7554/eLife.93117.4.sa1
Reviewer #2 (Public review): https://doi.org/10.7554/eLife.93117.4.sa2
Reviewer #3 (Public review): https://doi.org/10.7554/eLife.93117.4.sa3
Author response https://doi.org/10.7554/eLife.93117.4.sa4

---

# Additional files

### Supplementary files
• Supplementary file 1. Primers table used in this study.

• MDAR checklist

### Data availability
All data are available in the manuscript and supporting files; source data files have been provided for Figures 1–4 and their figure supplements (For Figure 1 and Figure 3). Source data 1 and Source data 2 files of figures and figure supplements contain original files of western blots.

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
