## [Editor Report · eLife Assessment]

This **important** manuscript demonstrates that UGGT1 is involved in preventing the premature degradation of endoplasmic reticulum (ER) glycoproteins through the re-glucosylation of their *N*-linked glycans following release from the calnexin/calreticulin lectins. The authors include a wealth of **convincing** data in support of their findings, although extending these findings to other types of substrates, such as secreted proteins, could further demonstrate the global importance of this mechanism for protein trafficking through the secretory pathway. This work will be of interest to scientists interested in ER protein quality control, proteostasis, and protein trafficking.

---

## [Referee Report · Reviewer #1 (Public review)]

Summary:

UGGTs are involved in the prevention of premature degradation for misfolded glycoproteins, by utilizing UGGT1-KO cells and a number of different ERAD substrates. They proposed a concept by which the fate of glycoproteins can be determined by a tug-of-war between UGGTs and EDEMs.

Strengths:

The authors provided a wealth of data to indicate that UGGT1 competes with EDEMs, which promotes the glycoprotein degradation.

---

## [Referee Report · Reviewer #2 (Public review)]

In this study, Ninagawa et al., sheds light on UGGT's role in ER quality control of glycoproteins. By utilizing UGGT1/UGGT2 DKO , they demonstrate that several model misfolded glycoproteins undergo early degradation. One such substrate is ATF6alpha where its premature degradation hampers the cell's ability to mount an ER stress response.

This study convincingly demonstrates that many unstable misfolded glycoproteins undergo accelerated degradation without UGGTs. Also, this study provides evidence of a "tug of war" model involving UGGTs (pulling glycoproteins to being refolded) and EDEMs (pulling glycoproteins to ERAD).

The study explores the physiological role of UGGT, particularly examining the impact of ATF6α in UGGT knockout cells' stress response. The authors further investigate the physiological consequences of accelerated ATF6α degradation, convincingly demonstrating that cells are sensitive to ER stress in the absence of UGGTs and unable to mount an adequate ER stress response.

These findings offer significant new insights into the ERAD field, highlighting UGGT1 as a crucial component in maintaining ER protein homeostasis. This represents a major advancement in our understanding of the field.

---

## [Referee Report · Reviewer #3 (Public review)]

This valuable manuscript demonstrates the long-held prediction that the glycosyltransferase UGGT slows degradation of endoplasmic reticulum (ER)-associated degradation substrates through a mechanism involving re-glucosylation of asparagine-linked glycans following release from the calnexin/calreticulin lectins. The evidence supporting this conclusion is solid using genetically-deficient cell models and well established biochemical methods to monitor the degradation of trafficking-incompetent ER-associated degradation substrates, although this could be improved by better defining of the importance of UGGT in the secretion of trafficking competent substrates. This work will be of specific interest to those interested in mechanistic aspects of ER protein quality control and protein secretion.

The authors have largely addressed my comments from the previous round of review. The only remaining comment is about defining the impact of UGGT1 in the regulation of secretion-competent proteins, which the authors indicate they will continue to pursue in subsequent work, which is fine, but remains a minor limitation of the study.

As I mentioned in my previous review, I think that this work is interesting and addresses an important gap in experimental evidence supporting a previously asserted dogma in the field. I do think that the authors would be better suited for highlighting the limitations of the study, as discussed above. Ultimately, though, this is an important addition to the literature.

---

## [Author Response]

The following is the authors’ response to the previous reviews.

**Public Reviews:**

**Reviewer #1 (Public review):**
Summary:UGGTs are involved in the prevention of premature degradation for misfolded glycoproteins, by utilizing UGGT1-KO cells and a number of different ERAD substrates. They proposed a concept by which the fate of glycoproteins can be determined by a tug-of-war between UGGTs and EDEMs.Strengths:The authors provided a wealth of data to indicate that UGGT1 competes with EDEMs, which promotes the glycoprotein degradation.Weaknesses:NA

We appreciate your comment.

**Reviewer #2 (Public review):**
In this study, Ninagawa et al., sheds light on UGGT's role in ER quality control of glycoproteins. By utilizing UGGT1/UGGT2 DKO , they demonstrate that several model misfolded glycoproteins undergo early degradation. One such substrate is ATF6alpha where its premature degradation hampers the cell's ability to mount an ER stress response.This study convincingly demonstrates that many unstable misfolded glycoproteins undergo accelerated degradation without UGGTs. Also, this study provides evidence of a "tug of war" model involving UGGTs (pulling glycoproteins to being refolded) and EDEMs (pulling glycoproteins to ERAD).The study explores the physiological role of UGGT, particularly examining the impact of ATF6α in UGGT knockout cells' stress response. The authors further investigate the physiological consequences of accelerated ATF6α degradation, convincingly demonstrating that cells are sensitive to ER stress in the absence of UGGTs and unable to mount an adequate ER stress response.These findings offer significant new insights into the ERAD field, highlighting UGGT1 as a crucial component in maintaining ER protein homeostasis. This represents a major advancement in our understanding of the field.

Thank you very much for your comment.

**Reviewer #3 (Public review):**
This valuable manuscript demonstrates the long-held prediction that the glycosyltransferase UGGT slows degradation of endoplasmic reticulum (ER)-associated degradation substrates through a mechanism involving re-glucosylation of asparaginelinked glycans following release from the calnexin/calreticulin lectins. The evidence supporting this conclusion is solid using genetically-deficient cell models and well established biochemical methods to monitor the degradation of trafficking-incompetent ER-associated degradation substrates, although this could be improved by better defining of the importance of UGGT in the secretion of trafficking competent substrates. This work will be of specific interest to those interested in mechanistic aspects of ER protein quality control and protein secretion.The authors have attempted to address my comments from the previous round of review, although some issues still remain. For example, the authors indicate that it is difficult to assess how UGGT1 influences degradation of secretion competent proteins, but this is not the case. This can be easily followed using metabolic labeling experiments, where you would get both the population of protein secreted and degraded under different conditions. Thus, I still feel that addressing the impact of UGGT1 depletion on the ER quality control for secretion competent protein remains an important point that could be better addressed in this work.

We mainly focused on the impact of UGGT1 depletion on ERAD in this paper and intend to determine the impact of UGGT1 depletion on the ER quality control for secretion competent protein in the near future.

Further, in the previous submission, the authors showed that UGGT2 depletion demonstrates a similar reduction of ATF6 activation to that observed for UGGT1 depletion, although UGGT2 depletion does not reduce ATF6 protein levels like what is observed upon UGGT1 depletion. In the revised manuscript, they largely remove the UGGT2 data and only highlight the UGGT1 depletion data. While they are somewhat careful in their discussion, the implication is that UGGT1 regulates ATF6 activity by controlling its stability. The fact that UGGT2 has a similar effect on activity, but not stability, indicates that these enzymes may have other roles not directly linked to ATF6 stability. It is important to include the UGGT2 data and explicitly highlight this point in the discussion. Its fine to state that figuring out this other function is outside the scope of this work but removing it does not seem appropriate.

We have added the data of UGGT2-KO and UGGT-DKO cells to Figure 4 and discussed appropriately.

As I mentioned in my previous review, I think that this work is interesting and addresses an important gap in experimental evidence supporting a previously asserted dogma in the field. I do think that the authors would be better suited for highlighting the limitations of the study, as discussed above. Ultimately, though, this is an important addition to the literature.

We appreciate your comments. Thank you very much.

**Recommendations for the authors:**

**Reviewer #1 (Recommendations for the authors):**
I have carefully gone through the revised manuscript and responses to the reviewers' comments; I believe that the authors did a great job on revisions, and I do think that now this manuscript has been much improved (far easier to read through). Now I have only minor comments as follows;Page 9: Lines 8-9; Comparison between WT and EDEM-TKO cells indicates that ATF6alpha is still degraded via gpERAD requiring mannose trimming even in the presence of DNJ (Fig. 1D). (it would be better to indicate which figure to look)

We have fixed it.

Page 10: Lines 9-11; as multiple higher molecular weight bands representing a mixture of G3M9, G2M9m and GM9 etc. in WT cells treated with CST -> I am NOT AT ALL convinced with this statement on Figure 1-figure supplement 6A. How can the subtle glycan structure difference cause the ladder of the band? And if it is indeed the case (which I frankly doubt by the way), will endo-alpha-mannosidase treatment end up with a single band for CST? And PNGase F digestion can cancel all size difference between samples (control, +DNJ and +CST)?

CD3d-DTM-HA is a small protein (~20 kDa) possessing three N-glycans. Clear increase in the level of GM9 in WT cells treated with DNJ (Figure 1-Figure supplement 5A) caused an upward band shift (Figure 1-Figure supplement 6A). Similarly, clear increase in the levels of GM9, G2M9, G3M9 in WT cells treated with CST (Figure 1-Figure supplement 6B) produced the ladder of the band (Figure 1-Figure supplement 6A).

Crystal violet assay (new Fig 4G; Page 33); It said that, after treating cells with drug (Tg) for 4 hours, cells were spread on 24 well plates and cultured without Tg for 5 days. If incubated that long, I wonder that any compromised viability may have been canceled by growing cells (cells become confluent no matter what?). Am I missing something? Please clarify.

We employed a previously published method to determine ER stress sensitivity (Yamamoto et al., Dev. Cell, 2007). Although any compromised viability may have been canceled by growing cells, as suggested, we were able to detect the difference between WT and UGGT-KO cells.

Figure 5D; why one of the three N-glycans is missing on the last protein??

We have fixed it.